# Association Study between Antioxidant Nutrient Intake and Low Bone Mineral Density with Oxidative Stress-Single Nucleotide Variants: *GPX1* (rs1050450 and rs17650792), *SOD2* (rs4880) and *CAT* (rs769217) in Mexican Women

**DOI:** 10.3390/antiox12122089

**Published:** 2023-12-08

**Authors:** Rogelio F. Jiménez-Ortega, Diana I. Aparicio-Bautista, Adriana Becerra-Cervera, Priscilla López-Montoya, Guadalupe León-Reyes, Jeny Flores-Morales, Manuel Castillejos-López, Alberto Hidalgo-Bravo, Jorge Salmerón, Berenice Rivera-Paredez, Rafael Velázquez-Cruz

**Affiliations:** 1Laboratorio de Genómica del Metabolismo Óseo, Instituto Nacional de Medicina Genómica (INMEGEN), Mexico City 14610, Mexico; rogeliofrank.jimenez@uneve.edu.mx (R.F.J.-O.); daparicio@inmegen.gob.mx (D.I.A.-B.); abecerra@inmegen.edu.mx (A.B.-C.); priscilla_lopez92@comunidad.unam.mx (P.L.-M.); greyes@inmegen.gob.mx (G.L.-R.); jfloresm@inmegen.gob.mx (J.F.-M.); 2Consejo Nacional de Humanidades, Ciencias y Tecnologías (CONAHCYT), Mexico City 03940, Mexico; 3Unidad de Epidemiología Hospitalaria e Infectología, Instituto Nacional de Enfermedades Respiratorias Ismael Cosio Villegas (INER), Mexico City 14080, Mexico; manuel.castillejos@iner.gob.mx; 4Departamento de Medicina Genómica, Instituto Nacional de Rehabilitación (INR), Mexico City 14389, Mexico; ahidalgo@inr.gob.mx; 5Centro de Investigación en Políticas, Población y Salud, Facultad de Medicina, Universidad Nacional Autónoma de México, Mexico City 04510, Mexico; drjorgesalmeron@unam.mx (J.S.); bereriverap@comunidad.unam.mx (B.R.-P.)

**Keywords:** bone mineral density, postmenopausal women, oxidative stress, SNVs, antioxidants intake

## Abstract

Oxidative stress is essential in developing multiple bone metabolism diseases, including osteoporosis. Single-nucleotide variants (SNVs) have been associated with oxidative stress, promoting an imbalance between the production of reactive oxygen species and the ability to neutralize them, and it has been reported that antioxidant nutrient intake can influence bone mineral density (BMD). This work reports the association between oxidative stress-related SNVs (*GPX1*-rs1050450, rs17650792, *SOD2*-rs4880, and *CAT*-rs769217), BMD, and antioxidant nutrient intake. The study included 1269 Mexican women from the Health Workers Cohort Study. Genotyping was performed using predesigned TaqMan assays. Dietary data were collected using a 116-item semi-quantitative food frequency questionnaire. A dietary antioxidant quality score (DAQS) was used to estimate antioxidant–nutrient intake. Association analysis was estimated via linear, logistic, or quantile regression models. The results showed an association of the rs1050450-A and rs17650792-A alleles with femoral neck BMD (*p* = 0.038 and *p* = 0.017, respectively) and the SNV rs4880-A allele with total hip BMD (*p* = 0.026) in respondents aged 45 years or older. In addition, antioxidant–nutrient intake was associated with the rs4880-GG genotype, being significant for fiber (*p* = 0.007), riboflavin (*p* = 0.005), vitamin B6 (*p* = 0.034), and vitamin D (*p* = 0.002). The study showed an association between oxidative stress-related SNVs, BMD, and antioxidant–nutrient intake in Mexican women. Therefore, treatments for low BMD could be developed based on antioxidant supplementation.

## 1. Introduction

Most organisms, including humans, live in an aerobic environment and are exposed to various oxidizing agents that are intentionally produced as by-products of metabolism. It has been reported that inflammation and oxidative stress affect vital biochemical reactions, generating reactive oxygen species (ROS) and impairing the cell’s antioxidant defense mechanisms. These events contribute to the development of different metabolic disorders such as type 2 diabetes (T2D), obesity, cardiovascular diseases, dyslipidemias, hypertension, and bone metabolism diseases such as osteoporosis (OP) [1,2]. The relationship between T2D and BMD variation is complex. In recent years, it has been reported that T2D affects the metabolism of carbohydrates, fatty acids, and proteins, causing deregulation of calcium, phosphorus, and magnesium, contributing to development of bone metabolism disorders, including osteoporosis [3].

OP is described as a microstructural degeneration of bone tissue and decreased bone mass, resulting in increased bone fragility and fracture susceptibility [4]. There is an increased incidence of osteoporotic fractures at all ages; however, when women reach menopause, they have twice the risk of fractures compared to men due to estrogen deficiency, which causes an increase in bone turnover [5]. OP is a multifactorial and complex disease determined by genetic and environmental factors. In recent years, it has been proposed that oxidative stress (OS) may contribute to the development of OP. In vitro and animal model studies have shown that OS may induce bone loss by regulating the activation of cells responsible for bone remodeling [6,7,8,9]. Antioxidant defense mechanisms protect cells from free radical damage. These systems include enzymes, such as superoxide dismutase (SOD), glutathione reductase (GR), glutathione peroxidase (GPX), and catalase (CAT); and minerals, such as selenium (Se) and zinc (Zn); in addition to non-enzymatic antioxidants or nutrient-derived antioxidants, such as vitamins A, C, and E [10]. 

Epidemiological studies have reported an inverse association between intake of antioxidant vitamins and carotenoids with low BMD, fracture risk, and OP [11,12,13]. Numerous factors can alter the efficacy of antioxidant enzymes, such as single-nucleotide variants (SNVs) in genes encoding redox enzymes. Variants in *CAT* (rs769217), *SOD2* (rs4880), and *GPX1* (rs1050450 and rs17650792) genes have been associated with defense against oxidative stress [14]. 

Recent studies have demonstrated the association between genetic variants in antioxidant genes with oxidative stress in different types of cancer. Nikic et al. (2023) demonstrated that there is an association between SNVs in *GPX1* (rs1050450) and *SOD2* (rs4880), with the risk of developing urogenital bladder cancer, independently and in combination with smoking. Additionally, rs1050450 is associated with the aggressiveness of the disease [15]. In another study, Djokic et al. (2022) analyzed the association of SNVs in the antioxidant genes *GPX1* (rs1050450) and *SOD2* (rs4880) and the transcription factor *Nrf2* (rs6721961) with the risk of prostate cancer in the Serbian population. Furthermore, they propose that these SNVs could be used as risk biomarkers for the evolution of prostate cancer [16]. Despite the importance of these antioxidant mechanisms, there is a lack of studies investigating the relationship between antioxidant nutrient intake and bone mineral density (BMD) status. This study aims to analyze the relationship between SNVs in oxidative stress-related genes with the variation of BMD and antioxidant nutrient intake in a population of Mexican women. Additionally, we evaluated the role of SNVs in oxidative stress-related genes associated with metabolic traits, such as glucose levels and diabetes status.

## 2. Materials and Methods

### 2.1. Study Population

The population sample consisted of a total of 1269 unrelated women over 18 years of age who were recruited from the second wave (2010–2012) of the “Health Worker Cohort Study (HWCS)” by the Mexican Social Security Institute (IMSS) in the city of Cuernavaca, Morelos (central Mexico); this is a long-term study focused on lifestyle and factors leading to the development of chronic diseases. Women born in Mexico whose parents and grandparents identified themselves as mestizo-Mexican were included in this study [17]. 

### 2.2. Bone Mineral Density Measurement (Assessment)

BMD was assessed at the total hip (TH), femoral neck (FN), and lumbar spine (LS) L2-L4 body sites via dual-energy X-ray absorptiometry (DXA) on a Lunar DPX NT (Lunar Radiation Corp., Madison, WI, USA). Measurements are expressed in grams per square centimeter (cm^2^). BMD data were analyzed according to T-score, using the criteria defined by the World Health Organization (WHO) [18]. Low BMD at any skeletal site (osteopenia/osteoporosis) was defined as T-score ≤ −1.0. The normal group (N) included women with T-scores above −1.0.

### 2.3. Anthropometric and Clinical Evaluation 

A venous blood sample was taken from each patient after eight hours of fasting, and each sample was separated via centrifugation to recover blood serum, which was used for biochemical assay. T2D was determined if the participants presented at least one of the following three criteria: (1) self-report T2D diagnosed by a physician, (2) use of hypoglycemic medication, and (3) fasting glucose ≥126 mg/dL [19]. Blood glucose was measured via an enzymatic method on a Selecta XL (Randon Laboratories Ltd., Antrim, UK) according to the manufacturer’s instructions and processed at the IMSS laboratory in Cuernavaca, Morelos, following standardized procedures according to the guidelines of the International Federation of Clinical Chemistry and Laboratory Medicine [20]. An age of 45 years or more was used as a proxy indicator of menopausal status, as seen in previous studies [21,22,23].

### 2.4. DNA Extraction and Genotyping Analysis 

Peripheral blood was collected from each participant in EDTA tubes stored at −20 °C until genomic DNA extraction. Total DNA was extracted using a commercial isolation kit (QIAGEN system Inc., Valencia, CA, USA) according to the manufacturer’s instructions. Genotyping was performed according to TaqMan^®^ SNP Genotyping Assays user guide shown in: https://assets.thermofisher.com/TFS-Assets/LSG/manuals/cms_040597.pdf (accessed on 15 August 2023), using the following predesigned TaqMan probes (Applied Biosystems, Massachusetts, MA, USA, EE.UU.): rs769217 (C___3102907_10) in the *CAT* gene, rs4880 (C___8709053_10) in *SOD2*, and rs1050450 and rs17650792 (C_175686987_10 and C___7911963_10, respectively) in *GPX1* [14]. Real-time PCR was performed on a Quant Studio 7 Flex (Applied Biosystems, Massachusetts, MA, USA, EE.UU.). Data were analyzed using Sequence Detection System (SDS) software, version 2.2.1.

### 2.5. Dietary Assessment

Dietary intake information was evaluated using a 116-item semi-quantitative food frequency questionnaire (FFQ); its validity and reliability have been previously reported [23,24]. The data were collected from a self-administered questionnaire, and the reported frequency for each food item was converted to a daily intake. Food composition tables, compiled by the National Institute of Public Health, were used to determine the nutrient compositions of all foods and the diet parameters were adjusted for energy using the residual method [25,26].

A dietary antioxidant quality score (DAQS) was used to calculate antioxidant–nutrient intake. These scores refer to the intake of certain vitamins and minerals that have been proven to act as dietary antioxidants, including selenium, zinc, magnesium, vitamin A, vitamin C, and vitamin E. Both indices were created following the methodology published previously [27]. We compared the daily nutrient intake of participants with the recommended daily intake guidelines specifically for the DAQS [28].

### 2.6. Statistical Analysis

Data are presented as median and percentile 25–75 for continuous variables and percentages for categorical variables. All variables were examined for normality. The chi-square test was used to evaluate Hardy–Weinberg equilibrium (HWE) for each SNV. To explore the associations of interest, we estimated linear regression or quantile regression models as appropriate. For biochemical variables such as glucose and lipids, we performed a logarithmic transformation. For binary variables, we estimated logistic regression models. The models were adjusted for age, body mass index (BMI), smoking status, and physical activity. The dietary parameters were adjusted for total energy intake using the residual method. All statistical assessments were two-sided and considered significant when the *p*-value was ≤0.05.

## 3. Results

### 3.1. Population Characteristics

The characteristics of the participants are shown in Appendix A. Our study population comprised 1269 women with a median age of 53 years (P25–P75: 42–62); most of them were women aged ≥45 years old (70.3%) (*p* < 0.001). The median BMI in the total women was 26.7 (23.9–30.0) kg/m^2^; more than half of them (75.3%) had overweight/obesity. We observed statistically significant differences by age group. Women ≥45 years old had a higher median BMI (*p* < 0.001), a higher prevalence of overweight/obesity (*p* = 0.0001)/(*p* = 0.001), a higher prevalence of diabetes (*p* < 0.001), elevated levels of triglycerides *(p* < 0.001), LDL (*p* < 0.001), and cholesterol (*p* < 0.001), lower BMD at different body sites (*p* < 0.001), and lower consumption of vitamin C (*p* < 0.001), omega-6 (*p* = 0.0007), and saturated fat (*p* < 0.001) compared to women <45 years old.

### 3.2. Allele Frequencies of SNVs

The allele frequencies of the four SNVs are reported in Appendix A. We compared the allele frequency of the SNVs with those of the populations included in the 1000 Genomes Project. The SNVs showed differences in allele frequencies in our cohort compared to the CEU population but were similar to data from the MXL population.

### 3.3. Association Analyses between Oxidative Stress-Related SNVs and Bone Mineral Density 

We explored the association between the SNVs in oxidative stress-related genes and BMD at different sites. We observed that individuals carrying at least one copy of the A allele of the rs17650792 variant had a lower median of femoral neck BMD (*p* = 0.0472) (Appendix A). In the adjusted models, a statistically significant association was found in the recessive model for total hip BMD (*p* = 0.031) and femoral neck BMD (*p* = 0.012) (Table 1). When stratified by age groups, we found that rs1050450 (codominant model AA = 0.025 and recessive *p* = 0.021) and rs17650792 (recessive *p* = 0.028) were associated with lower BMD at the total hip in women <45 years old. Conversely, in women aged 45 and older, rs769217 (*p_codominant_* = 0.007 and *p_dominant_* = 0.031) and rs4880 (*p_additive_* = 0.040, *p_dominant_* = 0.026 and *p_recessive_* = 0.039) were associated with higher total hip BMD (Appendix A). On the other hand, rs1050450 (*p_additive_* = 0.038) and rs17650792 (*p_additive_* = 0.017, *p_codominant_* = 0.002 and *p_recessive_* = 0.002) were linked to increased odds of having low femoral neck BMD (Table 2). Upon age stratification, the rs17650792 variant remained statistically significant for lower femoral neck BMD in both age groups (Appendix A). Furthermore, in women aged ≥45 years, rs4880 and rs769217 exhibited a protective effect against reduced total hip (*p_codominant_* = 0.026 and *p_recessive_* = 0.009) and lumbar spine BMD (*p_codominant_* = 0.035 and *p_dominant_* = 0.031, respectively (Appendix A).

### 3.4. Dietary Antioxidant Intake and Oxidative Stress-Related SNV Association

We observed that the A allele of rs1050450 was associated with a higher median intake of vitamin B12 and omega-6. These associations remained statistically significant in the adjusted models (Table 3 and Appendix A). Additionally, we observed that carrying at least one copy of the A allele of rs1050450and rs17650792 variants was associated with higher odds of having a high DAQS index consumption compared to women carrying the ancestral allele (*p_codominant_* = 0.003 and *p* = 0.04, respectively) (Table 3). Under a recessive model, we found that the rs769217-C variant was associated with higher selenium intake (*p* = 0.017) (Table 4). Finally, the rs4880-A variant was associated with a lower intake of retinol (*p* = 0.023) (Appendix A), vitamin E (*p_recessive_* = 0.043), and fiber (*p_recessive_* = 0.003 and *p_codominant_* = 0.007) and a higher intake of riboflavin (*p_codomiant_* = 0.005) and vitamin D (*p_codominant_* = 0.002 and *p_recessive_* = 0.002) (Table 5).

### 3.5. Association Analyses between Oxidative Stress-Related SNVs and Metabolic Traits

Individuals carrying at least one copy of the T allele of the rs769217 variant were associated with a higher median of glucose (*p* = 0.029) and total cholesterol (*p* = 0.045) and a higher prevalence of diabetes (*p* = 0.032) (Appendix A). The adjusted models remained statistically significant for glucose levels (*p_codominant_* = 0.008 and *p_recessive_* = 0.007) and T2D (*p_additive_* = 0.043) (Table 4). When stratified by age groups, the associations remained significant in women ≥45 years for glucose levels (additive model β = 0.02, 95% CI 0.002, 0.047, *p* = 0.037) and T2D (additive model OR = 1.42, 95% CI 1.09–1.85, *p* = 0.009). 

## 4. Discussion

Several studies suggest a tight association between oxidative stress and the pathogenesis of OP, which is considered a major public health problem worldwide. In this study, we explored the association between genetic variants in oxidative stress-related enzymes; rs1050450, rs17650792, rs769217, and rs4880; and their association with BMD and their effect on dietary antioxidant intake in Mexican women.

In our study, the SNV rs1050450-A was significantly associated with low BMD levels in the femoral neck and total hip. According to a study of Slovenian subjects, the C/C-rs1050450 genotype was significantly associated with high values of femoral neck and total hip BMD (*p* < 0.026 and *p* = 0.023, respectively). In contrast, the homozygous T/T genotype was associated with low BMD values in the lumbar spine and total hip (*p* = 0.032 and *p* = 0.018, respectively) [29]. These results were consistent with those obtained in this work, since an association was observed between the rs1050450 with femoral neck BMD (*p* = 0.050) and the low BMD in total hip in <45 years of age. This variant has been linked to multiple oxidative stress-related diseases, such as polycystic ovary syndrome [30], endometriosis-related infertility [31], metabolic syndrome [32], and breast cancer, where the T allele of the rs1050450 variant is associated with reduced enzyme activity [33]. In this context, a lower GPX antioxidant activity has been related to OP in Mexican subjects ≥60 years of age, which could suggest that oxidative stress is a risk factor for OP [34]. 

The findings of this study revealed an association between the SNV rs17650792 and a low femoral neck BMD (*p* = 0.047). Although this SNV has not previously been associated with BMD values, some studies have reported its association with the development of prostate cancer. The products of *GPX1* can modify selenium transport and concentration, counteracting oxidative stress [14,35,36]. In bone metabolism, osteoblasts produce *GPX* to protect against ROS; however, changes in ROS and/or antioxidant systems seem to be involved in the pathogenesis of bone loss, contributing to the development of OP [37]. 

On the other hand, it has been reported that the DAQs index can favorably influence bone health in women [1,38]. In our study, we observed that variants rs1050450 and rs17650792 of the *GPX1* are positively associated with the DAQs index. However, carriage of risk alleles was associated with a risk of low BMD; the intake of individual antioxidants was analyzed. The rs1050450 variant presented an association under additive model with the intake of vitamin B12 (*p* = 0.035) and omega 6 under dominant model (*p* = 0.026). A high intake of antioxidants has a beneficial effect on bone health; however, it has been reported that high intake omega 6 fatty acids is negatively associated with BMD in postmenopausal women, which could be deleterious for bone health [39]. It is possible that individuals carrying the rs1050450 and rs17650792 variants, to compensate for the oxidative stress caused by the risk variant, tend to consume more antioxidant-rich foods. However, given that these variants are associated with lower oxidative capacity and a higher risk of low BMD, this complex interplay of genetic and dietary factors could explain our observed results. Additional studies are necessary to clarify the effect of those nutrient patterns on BMD.

Another key enzyme representing a barrier against oxidative damage is the antioxidant enzyme CAT, which converts ROS and hydrogen peroxide into water and oxygen to mitigate the toxic effects of hydrogen peroxide [40]. Our study observed a protective effect of the rs769217 variant in *CAT* gene on bone density. However, in a Hungarian population, only male carriers of the rs769217-T had lower femur density, suggesting that this variant might be sex-specific [41]. In addition, we found an association of this variant with elevated glucose levels (*p* = 0.02) and an increased risk of T2D (*p* = 0.03), while it was also related to higher selenium intake (*p* = 0.017). Recent studies have observed that the rs769217 variant might be associated with a higher concentration of hydrogen peroxide, which could be considered a risk factor for developing T2D, mainly in postmenopausal women [42]. A possible explanation is that catalase deficiency can chronically increase hydrogen peroxide levels in pancreatic B cells, which are sensitive to the oxidation and production of insulin, which could contribute to developing T2D [43,44]. Additionally, it was observed that high consumption of antioxidant nutrients reduces the risk of skeletal fluorosis. It has been reported that selenium could have an antioxidant function by scavenging free radicals, repairing membrane damage, and reducing fluoride-induced apoptosis [45,46]. In addition, selenium could enhance the activity of antioxidant enzymes SOD and GPX, reduce the toxic effect of fluoride, improve liver function, and inhibit apoptosis to a certain extent, improving BMD levels [47]. The rs769217 variant is associated with elevated glucose levels and an increased risk of type 2 diabetes and is linked to higher selenium consumption. An explanation for these results is that individuals carrying the rs769217 variant may be proactively adjusting their diet and lifestyle choices, including higher selenium consumption, as a conscious measure to reduce their risk of diabetes. These findings suggest a complex interaction between genetics, diet, and antioxidant pathways in influencing oxidative stress and its impact on glucose metabolism. 

On the other hand, manganese superoxide dismutase 2 is encoded by the *SOD2* gene, whose function is to convert superoxide into hydrogen peroxide, which is subsequently eliminated via its conversion into water and molecular oxygen by other oxidizing enzymes: catalase and glutathione peroxidase [48]. Interestingly, our results revealed that the SNV rs4880 ancestral allele is linked to low total hip BMD (*p* = 0.036), indicating a protective effect conferred by the A allele (OR = 0.58; 95% CI, 0.35–0.96). In the same way, in a population of postmenopausal women from Asian India, the ancestral allele was also significantly associated with the risk of osteoporosis combined with high levels of oxidative stress; consequently, the presence of a higher frequency of allele C indicates the risk of disease [49]. These data were consistent with the results reported in this work, where we observed an association between rs4880 and low total hip BMD (*p* = 0.036), with an OR indicating a protective effect conferred by the A allele (OR = 0.58; 95% CI, 0.35–0.96) equivalent to the T allele reported in the previous study. For dietary parameters, we found an association between the rs4880-AA genotype with intake of fiber, riboflavin, vitamin B6, and vitamin D. It is worth noting that we did not directly assess the relationship between diet and BMD in our study. 

Our study’s observed association between the rs4880 variant and some antioxidant nutrient intake suggests that genetic factors may influence dietary choices. This association highlights the relevance of exploring potential mechanisms by which oxidative stress-related genetic variants may indirectly impact bone health via dietary patterns. However, it is essential to recognize that further research is needed to fully elucidate the underlying mechanisms and the extent of their contribution to the observed outcomes in terms of bone health.

The strengths of the current study are described below. First, this analysis included a large sample of women (n = 1269) compared to other observational studies. Second, this is the first effort to understand the effect of oxidative stress variants, dietary antioxidant intake, and BMD in Mexican women. This study has some limitations. First, total oxidant status was not evaluated, so we cannot determine the direct effect of these variants on the antioxidant activity of the enzymes. Furthermore, it was a cross-sectional study that only partially analyzed dietary antioxidant intake. The cross-sectional design limits our ability to establish causal relationships or capture minor changes over time. Therefore, comprehensive longitudinal analyses are warranted to provide a more in-depth understanding of the dynamic interplay between genetic variants, dietary factors, and bone mineral density. Second, the FFQ data were self-reported; this can be subject to random measurement errors, leading to underestimating the association. However, the FFQ was designed and validated by the National Institute of Nutrition in Mexico to assess long-term exposure to different nutrients to study their potential role as risk factors for chronic disease. Third, we did not adjust for multiple comparisons because of the size found and the limited statistical power. However, recent studies have observed that adjustment via multiple tests controls global type-I error but considerably inflates type-II error [50]. Fourth, while our analysis did not identify a direct mediation relationship, it is important to acknowledge the complexity of the interplay between genetic factors, dietary components, and bone health. Additional factors, such as gene–environment interactions and other lifestyle variables, could contribute to the observed associations.

Our analysis has limited statistical power; nonetheless, it sheds light on the interplay between genetic variants, dietary factors, and bone mineral density BMD in Mexican women. These data hold relevance as we represent a population with distinct genetic and lifestyle characteristics. While our findings provide valuable insights, we recognize the importance of further research with larger sample sizes to corroborate and expand upon these observations. Nevertheless, our study contributes to the growing body of knowledge regarding the influence of genetic variations and dietary choices on bone health in the Mexican population, which may inform future strategies for bone disease prevention and management.

## 5. Conclusions

The study showed a positive association between oxidative stress-related rs17650792 and rs1050450-A variants and low BMD despite high DAQs intake, whereas rs769217 and rs4880 variants showed a protective effect with BMD probably associated with increased antioxidant intake in Mexican women. Antioxidant supplementation may be beneficial in preventing bone loss; however, studies regarding oxidant status are needed to determine the impact of antioxidant supplements on BMD. Our findings highlight the importance of understanding the interaction between genetic variants, dietary intake, and bone health in the Mexican population, which could have significant implications for preventing and treating bone diseases. A longitudinal study to evaluate the interaction between dietary antioxidant intake and oxidative stress SNVs on bone health would be helpful to the population. Therefore, based on the result obtained in this study, a dietary antioxidant intake could be suggested as an alternative treatment for low BMD (osteopenia/osteoporosis). In the future, these personalized nutritional recommendations could reduce the burden of oxidative stress-related diseases in the demographic group.

## Figures and Tables

**Table 1 antioxidants-12-02089-t001:** Association of *GPX1*, *SOD2*, and *CAT* variants with BMD (g/cm^2^) in Mexican women.

			rs1050450 (*GPX1*)		rs17650792 (*GPX1*)		rs4880 (*SOD2*)		rs769217 (*CAT*)	
BMD Site	Model	Genotype	b (95%CI)	*p* Value	b (95%CI)	*p* Value	b (95%CI)	*p* Value	b (95%CI)	*p* Value
Total hip	Additive		−0.003 (−0.017, 0.010)	0.617	−0.001 (−0.012, 0.010)	0.806	0.007 (−0.003, 0.016)	0.181	0.0001 (−0.010, 0.011)	0.980
	Codominant	GG	0.0		0.0		0.0		0.0	
		GA	−0.0003 (−0.016, 0.016)	0.975	0.010 (−0.004, 0.024)	0.155	0.002 (−0.012, 0.016)	0.796	0.011 (−0.003, 0.025)	0.131
		AA	−0.022 (−0.069, 0.025)	0.359	−0.027 (−0.056, 0.001)	0.063	0.018 (−0.004, 0.039)	0.110	−0.017 (−0.043, 0.009)	0.196
	Dominant	GG	0.0		0.0		0.0		0.0	
		GA + AA	−0.002 (−0.017, 0.013)	0.791	0.005 (−0.009, 0.018)	0.475	0.005 (−0.008, 0.019)	0.449	0.006 (−0.007, 0.020)	0.341
	Recessive	GG + GA	0.0		0.0		0.0		0.0	
		AA	−0.022 (−0.069, 0.025)	0.359	−0.031 (−0.060,−0.003)	0.031	0.017 (−0.004, 0.037)	0.108	−0.022 (−0.047, 0.003)	0.085
Femoral neck	Additive		−0.006 (−0.019, 0.007)	0.355	−0.004 (−0.015, 0.007)	0.452	0.005 (−0.005, 0.014)	0.336	0.002 (−0.008, 0.013)	0.634
	Codominant	GG	0.0		0.0		0.0		0.0	
		GA	−0.004 (−0.019, 0.012)	0.642	0.007 (−0.006, 0.021)	0.294	0.004 (−0.010, 0.018)	0.579	0.012 (−0.001, 0.026)	0.080
		AA	−0.025 (−0.071, 0.021)	0.293	−0.033 (−0.061,−0.005)		0.010 (−0.010, 0.031)	0.346	−0.011 (−0.036, 0.014)	0.408
	Dominant	GG	0.0		0.0		0.0		0.0	
		GA + AA	−0.005 (−0.020, 0.009)	0.477	0.002 (−0.011, 0.015)	0.797	0.005 (−0.008, 0.018)	0.434	0.009 (−0.004, 0.022)	0.193
	Recessive	GG + GA	0.0		0.0		0.0		0.0	
		AA	−0.024 (−0.079, 0.022)	0.307	−0.035 (−0.063,−0.008)	0.012	0.008 (−0.012, 0.028)	0.423	−0.016 (−0.040, 0.008)	0.191

CI = Confidence Interval; BMD: Bone Mineral Density; *GPX1*: Glutation peroxidase 1; *SOD2*: Superoxide dismutase 2; *CAT*: Catalase. Model adjusted for age, BMI, total energy, smoking status, physical activity, alcohol intake, calcium intake, calcium supplements, THR, and vitamin D intake.

**Table 2 antioxidants-12-02089-t002:** Association of *GPX1*, *SOD2*, and *CAT* variants with low BMD in Mexican women.

			rs1050450		rs17650792		rs4880		rs769217	
BMD Site	Model	Genotype	OR (95%CI)	*p* Value	OR (95%CI)	*p* Value	OR (95%CI)	*p* Value	OR (95%CI)	*p* Value
Total hip	Additive		1.02 (0.74–1.41)	0.909	1.16 (0.89–1.50)	0.267	0.88 (0.70–1.11)	0.277	1.06 (0.83–1.36)	0.639
	Codominant	GG	1.0		1.0		1.0		1.0	
		GA	1.00 (0.69–1.44)	0.991	1.02 (0.73–1.41)	0.924	1.11 (0.80–1.53)	0.541	0.93 (0.67–1.29)	0.655
		AA	1.19 (0.34–4.16)	0.783	1.77 (0.91–3.45)	0.092	0.61 (0.36–1.05)	0.072	1.37 (0.77–2.43)	0.286
	Dominant	GG	1.0		1.0		1.0		1.0	
		GA + AA	1.01 (0.71–1.44)	0.962	1.09 (0.80–1.50)	0.569	0.99 (0.72–1.35)	0.928	0.99 (0.72–1.35)	0.939
	Recessive	GG + GA	1.0		1.0		1.0		1.0	
		AA	1.19 (0.34–4.15)	0.782	1.76 (0.92–3.39)	0.090	0.58 (0.35–0.96)	0.036	1.42 (0.81–2.47)	0.217
Femoral neck	Additive		1.36 (1.02–1.82)	0.038	1.34 (1.05–1.70)	0.017	0.89 (0.72–1.10)	0.288	0.93 (0.74–1.17)	0.552
	Codominant	GG	1.0		1.0		1.0		1.0	
		GA	1.34 (0.96–1.88)	0.090	1.10 (0.82–1.49)	0.523	0.83 (0.61–1.13)	0.232	0.88 (0.65–1.19)	0.411
		AA	2.01 (0.72–5.58)	0.182	2.79 (1.46–5.33)	0.002	0.85 (0.53–1.34)	0.477	0.95 (0.55–1.65)	0.863
	Dominant	GG	1.0		1.0		1.0		1.0	
		GA + AA	1.38 (0.99–1.92)	0.053	1.24 (0.93–1.66)	0.137	0.83 (0.63–1.11)	0.216	0.89 (0.67–1.19)	0.438
	Recessive	GG + GA	1.0		1.0		1.0		1.0	
		AA	1.88 (0.68–5.20)	0.226	2.69 (1.42–5.10)	0.002	0.93 (0.60–1.43)	0.741	1.01 (0.60–1.72)	0.968

OR = Odds Ratio; CI = Confidence Interval; *GPX1*: Glutation peroxidase 1; *SOD2*: Superoxide dismutase 2; *CAT*: Catalase. Model adjusted for age, BMI, total energy, smoking status, physical activity, alcohol intake, calcium intake, calcium supplements, THR, and vitamin D intake.

**Table 3 antioxidants-12-02089-t003:** Association between variants rs1050450 and rs17650792 in *GPX1* with different dietary parameters.

		rs1050450						rs17650792	
		Vitamin B12 (mg/day)		Omega 6 (g/day)		DAQS		DAQS	
Model	Genotype	Coefficient ^&^ (95%CI)	*p* Value	Coefficient ^&^ (95%CI)	*p* Value	OR (95%CI)	*p* Value	OR (95%CI)	*p* Value
Additive		0.16 (0.01, 0.31)	0.035	0.30 (−0.09, 0.69)	0.125	1.54 (1.11–2.13)	0.009	1.30 (1.00, 1.71)	0.054
Codominant	GG	0.0		0.0		1.0		1.0	
	GA	0.23 (0.06, 0.41)	0.01	0.50 (0.05, 0.96)	0.030	1.76 (1.21–2.56)	0.003	1.19 (0.85–1.66)	0.324
	AA	−0.32 (0.84, 0.20)	0.225	0.20 (−1.12, 1.52)	0.768	1.22 (0.38–3.87)	0.736	2.12 (1.03–4.38)	0.042
Dominant	GG	0.0		0.0		1.0		1.0	
	GA + AA	0.19 (0.02, 0.36)	0.029	0.49 (0.06, 0.93)	0.026	1.71 (1.19–2.46)	0.004	1.28 (0.92–1.77)	0.141
Recessive	GG + GA	0.0		0.0		1.0		1.0	
	AA	−0.36 (−0.89, 0.17)	0.181	0.11 (−1.18, 1.41)	0.864	1.07 (0.34–3.35)	0.913	1.98 (0.98–4.05)	0.058

OR = Odds Ratio; CI = Confidence Interval. Model adjusted by age, BMI, smoking status, and physical activity. ^&^ Results of the quantile regression at the 50th percentile.

**Table 4 antioxidants-12-02089-t004:** Association between the rs769217 variant in *CAT* and selenium intake, glucose levels, and type 2 diabetes.

		Selenium (µg/day)		Glucose levels (mg/dL)		Type 2 Diabetes	
Model	Genotype	Coefficient (95%CI) ^&^	*p* Value	β (95%CI)	*p* Value	OR (95%CI)	*p* Value
Additive		0.63 (−0.86, 2.11)	0.410	0.02 (−0.0006, 0.03)	0.058	1.29 (1.01–1.66)	0.043
Codominant	TT	0.0		0.0		1.0	
	TC	−1.24 (−3.1, 0.60)	0.186	0.003 (−0.02, 0.03)	0.813	1.24 (0.89–1.74)	0.208
	CC	3.43 (−0.06, 6.92)	0.054	0.06 (0.02, 0.10)	0.008	1.77 (0.99–3.14)	0.052
Dominant	TT	0.0		0.0		1.0	
	TC + CC	−0.74 (−2.54, 1.06)	0.416	0.01 (−0.01, 0.03)	0.33	1.31 (0.95–1.81)	0.098
Recessive	TT + TC	0.0		0.0		1.0	
	CC	4.21 (0.76, 7.66)	0.017	0.06 (0.02, 0.10)	0.007	1.59 (0.92–2.76)	0.098

OR = Odds Ratio; CI = Confidence Interval. Model adjusted by age, BMI, smoking status, and physical activity. ^&^ Results of the quantile regression at the 50th percentile.

**Table 5 antioxidants-12-02089-t005:** Association between the rs4880 variant in *SOD2* and different dietary parameters.

		Saturated fat (g/day)		Fiber (g/day)		Riboflavin (mg/day)		Vitamin B6 (mg/day)		Vitamin D (UI/day)		Vitamin E (mg/day)	
Model	Genotype	Coefficient (95%CI)	*p* Value	Coefficient (95%CI)	*p* Value	Coefficient (95%CI)	*p* Value	Coefficient (95%CI)	*p* Value	Coefficient (95%CI)	*p* Value	Coefficient (95%CI)	*p* Value
Additive		−0.25 (−0.69, 0.19)	0.27	−0.70 (−1.47, 0.07)	0.073	0.04 (0.007, 0.08)	0.021	−0.03 (−0.06, 0.01)	0.167	0.28 (0.04, 0.53)	0.025	−0.09 (−0.26, 0.08)	0.282
Codominant	GG	0.0		0.0		0.0		0.0		0.0		0.0	
	GA	−0.92 (−1.56–0.30)	0.004	−0.14 (−1.18, 0.90)	0.796	0.03 (−0.03, 0.08)	0.310	0.002 (−0.05, 0.06)	0.949	0.05 (−0.30, 0.40)	0.767	0.009 (−0.24, 0.26)	0.942
	AA	0.17 (−0.81–1.14)	0.740	−2.23 (−3.83–0.62)	0.007	0.12 (0.04, 0.20)	0.005	−0.10 (−0.18–0.007)	0.034	0.85 (0.31, 1.39)	0.002	−0.37 (−0.75, 0.02)	0.060
Dominant	GG	0.0		0.0		0.0		0.0		0.0		0.0	
	GA + AA	−0.69 (−1.29–0.09)	0.024	−0.44 (−1.47, 0.58)	0.394	0.04 (0.02, 0.09)	0.171	−0.02 (−0.07, 0.04)	0.508	0.11 (−0.21, 0.44)	0.491	−0.04 (−0.27, 0.19)	0.723
Recessive	GG + GA	0.0		0.0		0.0		0.0		0.0		0.0	
	AA	0.60 (−0.33, 1.53)	0.207	−2.27 (−3.75–0.79)	0.003	0.10 (0.02, 0.18)	0.012	−0.10 (−0.18–0.01)	0.024	0.79 (0.28, 1.30)	0.002	−0.37 (−0.74–0.01)	0.043

CI = Confidence Interval. Model adjusted by age, BMI, smoking status, and physical activity. Results of the quantile regression at the 50th percentile.

## Data Availability

The data presented in this study are available on request from the corresponding author, R.V.-C. The data are not publicly available due to their containing information that could compromise the privacy of research participants.

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
