# Peer review of "Association Study between Antioxidant Nutrient Intake and Low Bone Mineral Density with Oxidative Stress-Single Nucleotide Variants: GPX1 (rs1050450 and rs17650792), SOD2 (rs4880) and CAT (rs769217) in Mexican Women"

_antioxidants, 2023, doi:10.3390/antiox12122089_

Round 1

Reviewer 1 Report

Comments and Suggestions for Authors

This is an interesting article with high novelty. Some points should be addressed:

- The first paragraph of the introduction section should be split into two paragraphs at the line 51.

- The introduction section is a bit small. More information concerning previous studies on the topic of the article should be added as a separate paragraph.

- The authors should specify if they mesured glucose levels themselves or they retrieved them from medical records in Anthropometric and Clinical Evaluation section.

- In the section Dietary assessment, the authors should specify if the data derived from questionnaires are self-reported or face-toface interviews between participants and qualified personnel were performed.

- In section 3.1, p-values and the number of Table for each p-value should be reported.

- In section 3.3, the number of Table for each p-values a should be reported.

- In sections 3.4 and 3.5, again, p-values with the corresponding Tables should be added into brackets at the end of the relevant sentences.

- In the discussion section, paragraphs 3 and 4 begin with the same phrasing (On the other hand). Please, used another beggining phrasing for one of the two paragraphs.

- In the last paragraph of the discussion section, the authors should emphasize that not causality effects could be derived due to the cross-sectional design of their study.

- In the last paragraph of the discussion section, the authors should also ephasize as a limitation of their stady that there nay be recall bias due to self-reported data.

- At the end of the conclusion section 1-2 sentences should be added about what future study could be performed based on the novel data of their study.

- Most of the cited studies and references are quite outdated. The authors should enhrich their references cited into the test with more recent references from the last 4-5 years.

- Moderate English language editing is recommended

Comments on the Quality of English Language

Moderate English language editing is recommended.

Author Response

Reviewer 1

Comments and Suggestions for Authors

This is an interesting article with high novelty. Some points should be addressed:

- The first paragraph of the introduction section should be split into two paragraphs at the line 51.

Response: The change was made.

- The introduction section is a bit small. More information concerning previous studies on the topic of the article should be added as a separate paragraph.

Response: Thanks for your observation, a paragraph about the SNV association of antioxidant genes and related diseases was added.

“Recent studies have demonstrated the association between genetic variants in an-tioxidant genes with oxidative stress in different types of cancer. Nikic et al. (2023) demonstrated that there is an association between SNVs in GPX1 (rs1050450) and SOD2 (rs4880), with the risk of developing urogenital bladder cancer, independently and in combination with smoking. Additionally, rs1050450 is associated with the aggressiveness of the disease [15]. In another study, Djokic et al. (2022), analyzed the association of SNVs in the antioxidant genes GPX1 (rs1050450), SOD2 (rs4880) and the transcription factor Nrf2 (rs6721961) with the risk of prostate cancer in Serbian population. Furthermore, they propose that these SNVs could be used as risk biomarkers for the evolution of prostate cancer [16].”

- The authors should specify if they measured glucose levels themselves or they retrieved them from medical records in Anthropometric and Clinical Evaluation section.

Response: In the Anthropometric and clinical evaluation section a paragraph has been added to answer the reviewer's request.

“Blood glucose was measured by an enzymatic method on a Selecta XL (Randox Laboratories Ltd., Antrim, UK) according to the manufacturer's instructions and processed at the IMSS laboratory in Cuernavaca, Morelos, following standardized procedures by the guidelines of the International Federation of Clinical Chemistry and Laboratory Medicine [20].”

- In the section Dietary assessment, the authors should specify if the data derived from questionnaires are self-reported or face-to face interviews between participants and qualified personnel were performed.

Response: We thank the reviewer for the observation. The first paragraph has been modified to respond to those requested by the reviewer and give greater clarity to the reader.

“Dietary intake information was evaluated using a 116-item semi-quantitative food frequency questionnaire (FFQ); its validity and reliability have been previously reported [23-24]. The data were collected from a self-administered questionnaire, and the reported frequency for each food item was converted to a daily intake.”

- In section 3.1, p-values and the number of Table for each p-value should be reported.

Response: We appreciate your valuable observation. P values for overweight, obesity, triglycerides, and LDL were added to Supplementary Table 1, and the number of tables for each p-value were added in the text.

- In section 3.3, the number of Table for each p-values a should be reported.

Response: The number of tables for each p-value were added.

- In sections 3.4 and 3.5, again, p-values with the corresponding Tables should be added into brackets at the end of the relevant sentences.

Response: The p-values with the corresponding Tables were added.

- In the discussion section, paragraphs 3 and 4 begin with the same phrasing (On the other hand). Please, used another beginning phrasing for one of the two paragraphs.

Response: The paragraph 3 begin was changed “The findings of this study revealed an association between the SNV rs17650792 and a low femoral neck BMD (p=0.047).”

- In the last paragraph of the discussion section, the authors should emphasize that not causality effects could be derived due to the cross-sectional design of their study.

Response: We appreciate your suggestion, a paragraph with the requested change was added.

“The cross-sectional design limits our ability to establish causal relationships or capture minor changes over time. Therefore, comprehensive longitudinal analyses are warranted to provide a more in-depth understanding of the dynamic interplay between genetic variants, dietary factors and bone mineral density.”

- In the last paragraph of the discussion section, the authors should also emphasize as a limitation of their study that there may be recall bias due to self-reported data.

Response: We understand the reviewer’s concern, however, we consider that there is no recall bias because individuals do not know their disease status, therefore they are not going to respond differentially.

In the discussion section, there is a paragraph addressing random measurement error leading to an underestimation of the association.

- At the end of the conclusion section 1-2 sentences should be added about what future study could be performed based on the novel data of their study.

Response: At the end of the conclusion section, we add the following sentence:

“A longitudinal study to evaluate the interactions between dietary antioxidant intake and oxidative stress SNV on bone health would be helpful to the population. Therefore, based on the results obtained in this study, a dietary antioxidant intake could be suggested as an alternative treatment for low BMD (osteopenia/osteoporosis). In the future, these personalized nutritional recommendations could reduce the burden of oxidative stress-related diseases in this demographic group”. 

- Most of the cited studies and references are quite outdated. The authors should enrich their references cited into the test with more recent references from the last 4-5 years.

Response: We appreciate your suggestion. Some more updated references were added to the text. However, some references are the direct antecedent of what is explained in the text.

- Moderate English language editing is recommended

Response: Thank you for your suggestion. The manuscript has been reviewed by an English-speaking native.

Reviewer 2 Report

Comments and Suggestions for Authors

The authors report associations between single nucleotide variants (SNV)  related to oxidative stress and bone mineral density. They detected  an association of rs1050450 -A and rs17650792-A and SNV res4880-A allele with the BMD of the proximal femur.  Furthermore, they found a association between  the res4880 GG genotype and antioxidant-nutrient intakes.  

The paper adds information on the role of oxidative stress for bone health.  The relation between the SNV and nutritional intakes of antioxidants  raises the question wether the individuals compensate the effect of SNV by increasing intake of antioxidants. 

The discussion is written conclusive and emphasizes the caution in interpretation of statistical associations. 

The authors discuss also the drawbacks of the study such as missing correction for multiple testing leading to type 1 error. 

specific comment:  The authors should provide a literature reference for the DNA genotyping analyses

Author Response

Reviewer 2

Comments and Suggestions for Authors

The authors report associations between single nucleotide variants (SNV) related to oxidative stress and bone mineral density. They detected an association of rs1050450 -A and rs17650792-A and SNV res4880-A allele with the BMD of the proximal femur.

Furthermore, they found an association between the res4880 GG genotype and antioxidant-nutrient intakes.  

The paper adds information on the role of oxidative stress for bone health.  The relation between the SNV and nutritional intakes of antioxidants raises the question whether the individuals compensate the effect of SNV by increasing intake of antioxidants. 

The discussion is written conclusive and emphasizes the caution in interpretation of statistical associations. 

The authors discuss also the drawbacks of the study such as missing correction for multiple testing leading to type 1 error. 

Specific comment:  The authors should provide a literature reference for the DNA genotyping analyses

Response: We appreciate your suggestion. We added the literature reference in the Section DNA extraction and genotyping analysis. Since the references on which the technique is based are old, the link to the user guide for the technique was added to the manuscript.  Additionally, we added a reference about oxidative stress-related SNV.

[14] Pascual-Geler M, Robles-Fernandez I, Monteagudo C, Lopez-Guarnido O, Rodrigo L, Gálvez-Ontiveros Y, Cozar JM, Rivas A, Alvarez-Cubero MJ. Impact of oxidative stress SNPs and dietary antioxidant quality score on prostate cancer. Int J Food Sci Nutr. 2020;71(4):500-508.

Reviewer 3 Report

Comments and Suggestions for Authors

The authors present a candidate gene analysis of BMD and dietary measures in a population of menopausal versus non-menopausal women. A few points to consider:

-I understand you eventually control for T2D in table 4 analyses, but what is the rationale for this. It seems to come up unexpectedly in the methods section to be used in the analyses. Maybe a sentence or two what the overlap of your OP investigations and T2D in the introduction will assist with understanding its place in the methods and results (table 4).

-suggest clarifying in the first sentence of 3.4 that this is b12 and omega-6 intake

-What is the correlations of the intake of the antioxidants with BMD measures? Specifically, looking at those that correlated to SNP. 

-Should you consider conducting a mediator analysis here to understand what these associations are telling us? Are the SNP correlations to BMD mediated by the antioxidant intake or the SNP correlations to antioxidant intake independent of the associations to BMD? Teasing this out would be interesting for this paper.

Author Response

Comments and Suggestions for Authors

The authors present a candidate gene analysis of BMD and dietary measures in a population of menopausal versus non-menopausal women. A few points to consider:

1) I understand you eventually control for T2D in Table 4 analyses, but what is the rationale for this. It seems to come up unexpectedly in the methods section to be used in the analyses. Maybe a sentence or two what the overlap of your OP investigations and T2D in the introduction will assist with understanding its place in the methods and results (Table 4).

Response: Thank you for your thoughtful review and valuable feedback on our manuscript. We appreciate the opportunity to address your concerns and further clarify the mention of Type 2 Diabetes (T2D) in the Methods section and its connection to our study.

Upon revisiting the manuscript, we acknowledge an oversight of the clarity of our description. We did not control for T2D, neither conducted a stratified analyses by diabetes status. The reference to T2D in the Methods section was in the context of exploring genetic variants related to oxidative stress and assessing their association with glucose levels and diabetes.

We intended to emphasize the broad scope of our investigation, considering the relevance of oxidative stress in the context of osteoporosis (OP) and its connection to diabetes. We recognize the importance of providing a more explicit rationale for including T2D-related variables in our analyses.

Considering this, we revised the manuscript to accurately reflect that T2D was not a covariate in the analyses. Instead, our focus was exploring genetic variants associated with oxidative stress, and we examined their relationships with glucose levels and the presence of diabetes (Table 4).

In the introduction, we added a paragraph on the relationship between T2D and BMD. “The relationship between T2D and BMD variation is complex. In recent years, it has been reported that T2D affects the metabolism of carbohydrates, fatty acids and proteins, causing deregulation of calcium, phosphorus and magnesium, contributing to development of bone metabolism disorders, including osteoporosis [3].”

We have incorporated into the objective the following sentence:

“Additionally, we evaluated the role of SNVs in oxidative stress-related genes associated with metabolic traits, such as glucose levels and diabetes status.” 

In addition, we would like to draw attention to a specific paragraph in the introduction that explicitly discusses the association of oxidative stress with diabetes. This acknowledgment further supports our rationale for exploring genetic variants related to oxidative stress in the context of osteoporosis and considering their potential implications for metabolic disorders such as type 2 diabetes (T2D). We hope this clarification provides a more comprehensive understanding of the background and objectives of our study.

Introduction paragraph:

“Most organisms, including humans, live in an aerobic environment and are exposed to various oxidizing agents that are intentionally produced as by-products of metabolism. It has been reported that inflammation and oxidative stress affect vital biochemical reactions, generating reactive oxygen species (ROS) and impairing the cell’s antioxidant defense mechanisms. These events contribute to the development of different metabolic disorders such as type 2 diabetes (T2D), obesity, cardiovascular diseases, dyslipidemias, hypertension, and bone metabolism diseases such as osteoporosis (OP) [1,2].”

2) suggest clarifying in the first sentence of 3.4 that this is b12 and omega-6 intake

Response: We appreciate your suggestion to clarify the first sentence in Section 3.4 regarding vitamin B12 and omega-6 intake. We have revised the text to explicitly state that this section explores the association between the A allele of rs1050450 and vitamin B12 and omega-6 intake. The modified text is as follows:

“We observed that the A allele of rs1050450 was associated with a higher median intake of vitamin B12 and omega-6. These associations remained statistically significant in the adjusted models (Table 3 and Supplementary Table 3).”

3) What is the correlations of the intake of the antioxidants with BMD measures? Specifically, looking at those that correlated to SNP. 

Response: Thank you for suggesting regarding estimating correlations of antioxidant intake with bone mineral density (BMD) measures.

Upon conducting Spearman correlations, we observed low correlation coefficients ranging from -0.075 to 0.093. While some of these correlations reached statistical significance, the consistent low values suggest a limited, and potentially weak, association between antioxidant intake and BMD measures in our study population.  

Recognizing the complex relationship between dietary factors and bone health is crucial. Several factors could contribute to the observed correlations, including the multifaceted nature of dietary patterns, genetic variations, and other lifestyle aspects.

Considering these findings, we recognize the need for further exploration and consideration of additional factors that may influence on the relationship between antioxidant intake and BMD. Additionally, this is a cross-sectional analysis, and the nature of such analyses may limit the ability to establish significant associations.

Furthermore, it's noteworthy to consider that in a previous longitudinal analysis published in 2022, we reported an association between a proinflammatory diet and bone mineral density. Therefore, we recommend conducting longitudinal analyses to reduce measurement error and provide a more comprehensive understanding of the dynamics between antioxidant intake and BMD over time.

Taking in count the consistent low correlation coefficients and the potential limitations of cross-sectional analyses, we are cautious about the interpretation and clinical significance of these results. Given the complex relationship between antioxidant intake and BMD, the observed correlations may not provide substantial insights into the broad context of bone health in our study population.

Therefore, after careful consideration, we propose that incorporating the specific results of these correlations into the manuscript may offer little value and may even risk misleading of the readers. Instead, we focused on the comprehensive discussion of the study limitations, the intricate interplay between various factors, and the importance of future longitudinal analyses.

Limitations paragraph:

“The cross-sectional design limits our ability to establish causal relationships or capture changes over time. Therefore, comprehensive longitudinal analyses are warranted to provide a more in-depth understanding of the dynamic interplay between genetic variants, dietary factors, and bone mineral density.”

Reference

[23]. Rivera-Paredez B, Quezada-Sánchez AD, Robles-Rivera K, Hidalgo-Bravo A, Denova-Gutiérrez E, León-Reyes G, Flores YN, Salmerón J, Velázquez-Cruz R. Dietary inflammatory index and bone mineral density in Mexican population. Osteoporos Int. 2022 Sep;33(9):1969-1979. doi: 10.1007/s00198-022-06434-7. Epub 2022 May 28. PMID: 35624319.

 Table.  Spearman correlation analysis between dietary nutrients and BMD

BMD total hip (g/cm2)

BMD femoral neck (g/cm2)

BMD lumbar spine (g/cm2)

Nutrients

rho

P value

rho

P value

rho

P value

Folate (µg/day)

-0.075

0.009

-0.065

0.024

-0.051

0.076

Retinol (UI/day

-0.058

0.047

-0.068

0.019

-0.042

0.152

Selenium (µg/day)

0.070

0.016

0.093

0.001

0.089

0.002

Vitamin C (mg/day)

-0.075

0.010

-0.094

0.001

-0.063

0.029

Vitamin E (µg/day)

-0.058

0.046

-0.065

0.025

-0.014

0.638

Zinc (mg/day)

-0.027

0.348

-0.008

0.794

-0.044

0.125

Magnesium (mg/day)

-0.094

0.001

-0.112

0.0001

-0.105

0.0003

DAQS

0.029

0.321

0.049

0.092

0.060

0.039

Vitamin B12 (mg/day)

-0.063

0.031

-0.034

0.240

-0.043

0.137

Omega 6 (g/day)

0.027

0.360

0.030

0.300

0.067

0.021

Saturated fat (g/day)

0.034

0.240

0.070

0.015

0.083

0.004

Fiber (g/day)

-0.042

0.152

-0.068

0.019

-0.071

0.015

Riboflavin (mg/day)

-0.100

0.0006

-0.067

0.021

-0.071

0.014

Vitamin B6 (mg/day)

-0.029

0.325

-0.026

0.363

-0.026

0.375

Vitamin D (UI/day)

-0.090

0.002

-0.061

0.034

-0.063

0.029

Vitamin E (µg/day)

-0.058

0.046

-0.065

0.025

-0.014

0.638

4) Should you consider conducting a mediator analysis here to understand what these associations are telling us? Are the SNP correlations to BMD mediated by the antioxidant intake or the SNP correlations to antioxidant intake independent of the associations to BMD? Teasing this out would be interesting for this paper.

Response: We examined the data to explore potential mediator effects in the association between SNP correlations, antioxidant intake and BMD. However, the results did not reveal statistically significant mediation effects.

 While our analysis did not identify a direct mediation relationship, it is important to acknowledge the complex interplay between genetic factors, dietary components, and bone health. Additional factors, such as gene-environment interactions and other lifestyle variables, could contribute to the observed associations. Although we did not find evidence of mediation in the current dataset, we recognize the value of further investigations, potentially with larger sample sizes, to comprehensively understand the intricate relationships among genetic variants, antioxidant intake and BMD.

We recognize this limitation that we did not observe a mediation effect in this study, will be incorporated into the discussion in the revised manuscript to enhance transparency and provide a comprehensive interpretation of our findings.

Additionally, the initial analysis revealed relatively low effect sizes for the association between SNPs and BMD, which could have an impact on the power to detect significant mediation effects. Furthermore, the correlations between dietary antioxidant intake and BMD measures were modest. Collectively, these factors suggest that the observed associations may be influenced by the inherent complexities of the relationships being investigated.

Round 2

Reviewer 1 Report

Comments and Suggestions for Authors

The revised form of the manuscript has significantly been improved.

Comments on the Quality of English Language

Minor editing of English language required